# Accuracy and Safety of Pedicle Screw Placement for Treating Adolescent Idiopathic Scoliosis: A Narrative Review Comparing Available Techniques

**DOI:** 10.3390/diagnostics13142402

**Published:** 2023-07-18

**Authors:** Alexandre Ansorge, Vishal Sarwahi, Ludmilla Bazin, Oscar Vazquez, Giacomo De Marco, Romain Dayer

**Affiliations:** 1Department of Spine Surgery, Lucerne Cantonal Hospital, 6000 Lucerne, Switzerland; 2Department of Pediatric Orthopedics, Cohen Children’s Medical Center, Northwell Health System, New Hyde Park, NY 11040, USA; 3Pediatric Orthopedic Unit, Geneva University Hospital, 1211 Geneva, Switzerland

**Keywords:** pedicle screw, adolescent idiopathic scoliosis, CT navigation, robotic-assisted, fluoroscopy-guided, free-hand

## Abstract

Posterior spinal fusion and segmental spinal instrumentation using pedicle screws (PS) is the most used procedure to correct adolescent idiopathic scoliosis. Computed navigation, robotic navigation, and patient-specific drill templates are available, besides the first described free-hand technique. None of these techniques are recognized as the gold standard. This review compares the PS placement accuracy and misplacement-related complication rates achieved with the techniques mentioned above. It further reports PS accuracy classifications and anatomic PS misplacement risk factors. The literature suggests a higher PS placement accuracy for robotic relative to computed navigation and for the latter relative to the free-hand technique (misplacement rates: 0.4–7.2% versus 1.9–11% versus 1.5–50.7%) using variable accuracy classifications. The reported PS-misplacement-related complication rates are, however, uniformly low (0–1.4%) for every technique, while robotic and computed navigation induce a roughly fourfold increase in the patient’s intraoperative radiation exposure relative to the free-hand technique with fluoroscopic implant positioning control. The authors, therefore, recommend dedicating robotic and computed navigation for complex deformities or revisions with altered landmarks, underline the need for a generally accepted PS accuracy classification, and advise against PS placement in grade 4 pedicles yielding higher misplacement rates (22.2–31.5%).

## 1. Introduction

PSF and SSI using pedicle screws through a single posterior approach is the most commonly used surgical technique to treat adolescent idiopathic scoliosis (AIS) patients. Basques et al. even found it the second most frequently performed pediatric orthopedic surgical procedure in the USA [1]. The rationale behind the popularity of segmental PS constructs is that, in comparison to earlier-developed procedures like the single Harrington rod constructs, the Luque sublaminar wire constructs and the hook constructs, it allows for better coronal and sagittal deformity correction; improves chest wall deformity correction, making thoracoplasty unnecessary for most of the cases; obviates the need for anterior release even for severe deformity cases; reduces the number of levels fused; and lowers the rate of implant-related surgical revisions [2,3,4,5,6]. 

Despite these recognized advantages, there are also concerns about using PS constructs related to potential screw misplacements and associated neurologic, vascular, and visceral complications that might even be lethal. The literature primarily focusses on the overall rate of screw misplacements to define the safety of PS instrumentation. In 2011, the Scoliosis Research Society and the Pediatric Orthopaedic Society of North America task force stated that screw placement accuracy might be a surrogate for complications and safety data. In a systematic review, they reported an overall PS accuracy rate of 94.9%, considering 17 studies, including 13,536 PS placed in 1353 pediatric patients, of whom the majority had AIS [7]. They did not analyze the used technique for PS placement in the included studies nor report the potential clinical impact of screw misplacement.

Currently, there is neither general agreement on the optimal technique for PS placement for treating AIS patients nor the definition of PS misplacement. The present narrative review therefore aims to present and compare relevant classification systems and assess PS placement accuracy; assess the clinical relevance of screw misplacements; discuss risk factors associated with PS misplacement; and present the available PS placement techniques, including free-hand with fluoroscopic assistance, computed navigation using 3D fluoroscopy, an O-arm (Medtronic, Dublin, Ireland) or a multidetector computed tomography (CT), robotic-assisted navigation techniques, and patient-specific drill templates with pre-planned trajectory, and compare their benefits and disadvantages in terms of accuracy and safety, including patient radiation exposure, and additionally describe the authors’ preferred PS placement technique.

## 2. PS Placement Accuracy Classifications

There is no generally accepted standardized method for determining the PS placement accuracy or defining PS misplacements. In 2007, Kosmopoulos and Schizas published a meta-analysis regarding PS placement accuracy [8]. They surprisingly identified 35 different PS placement assessment methods relying either on the analysis of radiographs or CTs, respectively, on dissection for the included cadaveric studies within the 130 included articles. More recently, Aoude et al. published a systematic review on the same subject and found less heterogenic results [9]. Out of the 68 included articles, 37 (54%) had a comparable methodology, representing the most used method for PS placement accuracy assessment. In these 37 articles, including nine articles considering AIS cases, PS accuracy was assessed using CT, and pedicle breaches were graded based on 2 mm increments. These grading systems do not consider the location of the breach or the remaining distance between the PS and the critical anatomical structures at risk. Most articles in this review consider breaches up to 2 mm as safe or acceptable and breaches over 2 mm as unsafe. The second most used grading system, which was reported in 16 articles (24%), is binary and classifies the PS either as “in” or “out”. Some studies defined PS to be “in” if the screw is completely contained in the pedicle; others allowed up to 25% of the screw diameter to breach the pedicle. 

Although simple to use with an intraoperative or postoperative CT, the most widely reported PS accuracy grading systems relying either on 2 mm breach increments or on the differentiation of screws being “in” or “out” fail to point out the clinical relevance of the breaches, which should actually be primarily focused on. This shortcoming was addressed by Sarwahi et al., who reported a novel classification system aimed at recognizing potentially clinically significant PS misplacements and considering the location of the breach and the remaining distance between the PS and the anatomical structures of concern [10]. This PS accuracy classification system defines the following gradings: accurately placed (AP); benign misplacement (BMP); indeterminate misplacement (IMP); and screw at risk (SAR). AP screws are, per definition, completely contained by the pedicle. BMPs breach the cortical wall but do not place any structure at risk; IMPs breach either medially by 2–4 mm or laterally respectively anteriorly, while >1–2 mm distance is kept between the screw tip and the anatomic structures at risk; and SARs either breach medially more than 4 mm or breach laterally or anteriorly and impinge (<1 mm distance between the screw tip and organ) on anatomic structures of concern, such as the aorta, the trachea, or the esophagus. The latter classification integrates the facts that medial breaches of up to 2 mm are generally accepted as safe and that transient or permanent neurologic deficit occurrence has been exclusively reported for medial breaches of at least 4 mm [8,9,10,11,12]. It also considers the risk associated with anterolaterally misplaced screws, as the definition of SAR encompasses PS, with an anterolateral breach impinging on an anatomic structure of concern. However, it is unknown if the chosen cut-off of <1 mm between the screw tip and the organ at risk for defining SAR due to anterolateral breaching is optimal. Foxx et al. gave some insights into the risk of vascular complication due to PS misplacement by reporting 9 cases summarizing 33 screws being in contact with a great vessel (aorta, iliac artery, iliac vein) within their studied cohort of 182 cases having undergone posterior spinal fusion (surgical indication not mentioned) [13]. Despite 33 SARs in their series, no vascular injury, including pseudoaneurysm, happened during the 9-year follow-up period. The analysis of this case series is reassuring; however, due to the relatively small size of the studied population and the presence of some case reports describing pseudoaneurysm formation or erosion of the aorta secondary to impinging PS, the risk of dramatic vascular complication occurrence is real and cannot be ignored [14,15]. 

In summary, a standardized method to assess the PS placement accuracy is needed to gain more evidence about the PS safety. The optimal classification system should be reproducible and simple and point out the clinical relevance of the breach. Because of the advantages of Sarwahi’s accuracy classification system presented here above, this method might fulfill the requirements for a standardized assessment method to be generally used in the future, if its reproducibility proves to be high and if the chosen cut-off (<1 mm distance between screw tip and organ) to define anterolaterally misplaced SAR proves to be clinically relevant. 

## 3. Patient-Related Risk Factors for Screw Misplacement

Liljenqvist et al. first reported the morphology of pedicles in AIS patients [16]. They analyzed 337 pedicles in 29 surgically treated AIS patients with CT and observed that the endosteal transverse pedicle width was significantly smaller in the apical region of thoracic curves (T7–T10 and T12) on the concave side (mean 2.5–5.2 mm) in comparison to the convex side (mean 4.1–5.9 mm). They, therefore, considered that PS instrumentation on the concavity in the apical region of thoracic curves is critical. Watanabe et al. stated that these imaging findings were not correlated to the practical feasibility of navigating a probe down a thoracic pedicle into the vertebral body and consequently reported a pedicle channel classification describing the osseous anatomy experienced during free-hand probe insertion into the pedicle channel in a prospective series of 53 consecutive scoliosis patients (including 38 AIS cases) [17]. They defined the following four types of pedicle channels: “Type A”, a pedicle probe smoothly inserted without difficulty, the morphology described as a “Large Cancellous Channel”; “Type B”, a pedicle probe inserted snugly with increased force, described as a “Small Cancellous Channel”; ”Type C”, a pedicle probe cannot be manually pushed but must be tapped with a mallet down the pedicle into the body, described as a “Cortical Channel”; and “Type D”, a pedicle probe cannot locate a channel, thus necessitating a “juxtapedicular” screw position, described as a “Slit/Absent Channel”. The distribution of pedicle channel types varied according to the diagnosis leading to scoliosis and was as follows for the included 38 AIS cases operated using 474 PS: 254 PS (53.6%) Type A, 157 PS (33.1%) Type B, 39 PS (8.2%) Type C, and 24 PS (5.1%) Type D. Thus, 87% of the pedicles (Types A and B) had cancellous channels, and 13% (Types C and D) had none. Watanabe et al. considered pedicles with a cancellous channel as relatively safe for free-hand probe insertion and PS instrumentation, and those without as needing more challenging techniques such as tapping the probe with a mallet, using a small high-speed drill or burr to perform a juxtapedicular placement. This prospective clinical channel classification was further correlated to the pedicle size measured in preoperative CT by subset data analysis to help surgeons preoperatively foresee PS placement technical challenges based on preoperative CT performance. However, Watanabe et al. did not correlate the pedicle channel grades to a corresponding PS placement success rate. Akazawa et al. revised the hereabove-presented channel grading system and defined the following grades based on preoperative CT measurements: grade 1 for a “large cancellous channel” with an inner diameter of ≥4 mm; grade 2 for a “moderate cancellous channel” with an inner diameter of ≥2 mm and < 4 mm; grade 3 for a “small cancellous channel” with an inner diameter of ≥1 mm and <2 mm; and grade 4 for a “cortical channel” with an inner diameter of <1 mm [18]. They used this revised pedicle channel grading system to describe 810 pedicles that were planned for PS placement and were probed for AIS correction in a series of 55 patients using an O-arm-based intraoperative navigation, a technology which basically consists of two units: a motorized circular fluoroscopy, able to rotate 360° around the region of interest to generate a series of projections used to generate 3D CT-like volume datasets based on the acquired 2D projections; and an additional navigation unit holding two infrared cameras used for matching and to localize the navigated instruments by geometrical triangulation. Azakawa et al. found that a higher pedicle channel grade was a significant risk factor for PS placement failure (significant difference for each increment). PS placement was considered a failure if perforation while probing occurred and was followed by PS placement discontinuation, if the PS was intraoperatively removed due to malposition on imaging, or if a PS deviation of at least 2 mm was seen in postoperative CT. There were 61 failures among the 810 probed pedicle channels. The respective failure rate breakdown was as follows for grades 1 to 4: 0.5%, 2.9%, 12%, and 31.5% (*p* < 0.001). They concluded that PS placement should be avoided in grade 4 pedicles (<1 mm inner diameter). Interestingly, although their series included 89 grade 4 pedicles (number of concerned patients unknown) with a failure rate of 31.5% for PS placement, no patient had neurologic or vascular complications. Another study by Akazawa et al. reported a significant PS misplacement rate increase for grade 4 pedicles (22.2%, *p* = 0.008) relative to other grades when using robotic navigation to treat AIS [19]. 

## 4. PS-Misplacement-Related Morbidity and Mortality

The literature mainly focusses on the overall rate of PS misplacements to define the safety of PS. The Scoliosis Research Society and the Pediatric Orthopaedic Society of North America task force have correspondingly stated that the PS placement accuracy might be considered as a surrogate for complication and safety data [7]. However, as previously reported, the overall rate of PS misplacement primarily measures the surgeons’ technical skills and is an unsatisfactory indicator of patient safety [11]. This manner of reporting PS misplacements indeed underestimates the patient’s incurred risk, as the overall rate of misplaced screws is much lower than the rate of concerned patients. This is particularly true for deformity surgeries, as the number of used PS per patient is high and as the deformity renders the PS placement more complex. The discrepancy between the overall and the per-patient rate of PS misplacement is well-illustrated by a series of 127 pediatric patients (89 with AIS) reported by Sarwahi et al., in which a total of 2724 PS were placed [11]. They had an overall PS misplacement rate of 1.1%, a per-patient PS misplacement rate of 14.2% when using their definition of SAR to define misplaced PS. In their series, 18 patients had SAR. Among them, 10 had medial misplacements, 6 had PS impinging on the aorta, 1 had screws impinging on the trachea, and 1 had a medial misplaced screw and another screw impinging on the aorta. Two patients had transient neurologic injury, which needed intraoperative PS removal due to loss of signals and a later surgical revision after neurological recovery. All PS impinging on the aorta were asymptomatic. Additionally, Suk et al. reported a series of 462 spinal deformity cases, including 330 idiopathic scoliosis patients corrected using a total of 4604 thoracic PS [20]. They found an overall PS misplacement rate of 1.5% and a per-patient PS misplacement rate of 10.4%, leading to one transient paraparesis and three dural tears. However, no significant PS-related neurologic or visceral complication adversely affected the long-tern outcome. In their systematic review concerning PS fixation complications in scoliosis surgery, Hicks et al. reviewed 21 studies totalizing 14,570 PS implanted in 1666 pediatric patients and found an overall PS misplacement rate of 4.3%, while the rate of patients with misplaced PS, which was reported only in a minority of the included studies, was about 11% [21]. The average revision rate for PS misplacement was 0.8% when considering 11 studies, including a total of 1436 patients. One pulmonary effusion resulting from an intrathoracic misplaced PS was reported. Four studies reported dural leaks during PS placement and found an incidental durotomy rate of 0.4% per inserted PS. The only reported neurologic complication was a transient paraparesis due to an epidural haematoma secondary to a medially misplaced PS [20]. No irreversible neurologic complications and no major vascular complications were recorded. 

From these case series, it can be assumed that PS misplacements occur at least in 10% of the pediatric deformity patients and that despite this high frequency, at least the short-term related morbidity is low. Additionally, even though the PS-misplacement-related mortality rate is unknown, it must be low, as the total (any cause) mortality rate for AIS cases was found to be 2.8 deaths per 1000 cases within 60 days of surgery in a large review based on the Scoliosis Research Society Morbidity and Mortality database [22]. However, the exact clinical relevance of PS misplacements remains unclear because most of them are at least initially asymptomatic, and little is known about their long-term natural history. 

## 5. Comparison of PS Placement Accuracy According to the Used Surgical Technique

The current literature lacks large-scale strong evidence-level studies comparing the PS placement accuracy obtained using free-hand, computed tomography, or O-arm, 3D fluoroscopy-based, or robotic navigation for PS instrumentation in pediatric deformity patients. The results of the relevant available reports presented in Table 1 and discussed below still suggest some higher PS placement accuracy for robotic relative to computed navigation and the latter relative to the free-hand technique. 

Concerning the accuracy of free-hand PS placement in pediatric deformity patients with intraoperative fluoroscopic implant positioning control, the available literature mainly consists of case series of various size and of some small comparative series comparing it to the use of computed navigation. The reported PS misplacement rates vary substantially (1.5–50.7%) [20,23,24,25,26,27,28]. This disparity might largely be attributed to inconsistent PS misplacement definitions and either postoperative X-rays or CT scans used for PS accuracy evaluation. In contrast to the considerable variation of the reported PS misplacement rates, the PS-misplacement-related complications were consistently low. No vascular complications and neurologic complications were barely reported at rates between 0% and 1.4% [20,23,24,25,26,27,28]. The most serious reported complications were transient paraparesis. The first large case series reporting free-hand PS instrumentation of the thoracic spine in pediatric deformity patients was published by Suk et al. in 2001 [20]. It included 462 patients (330 idiopathic scoliosis cases). Among the 4604 implanted PS, 67 were misplaced (1.5%) and 48 patients (10.4%) had misplaced PS. As the PS accuracy assessment relied on routine postoperative X-rays, and CT scans (*n* = 20) were only performed in case of clinical suspicion of mechanical complications or when the postoperative X-rays showed suspicious screws, PS misplacements might have been underreported. Also, the definition of PS misplacement was only qualitative (medial, lateral, superior, inferior). While the PS misplacements were potentially underreported, the reported safety remains substantial, as only one patient (0.2% of the patients) had a PS-misplacement-related complication and needed revision surgery. This patient had a medially misplaced PS, which caused an epidural haematoma with a transient paraparesis which resolved 3 weeks after PS removal and laminectomy. Exclusively, one RCT investigating PS placement accuracy compared the free-hand technique to computed navigation using 3D fluoroscopy [32], a technology based on a motorized C-arm that produces a series of projections by performing an incomplete rotation (approximately 190°) around an isocenter (region of interest) and can generate 3D CT-like volume datasets on the basis of the acquired 2D projections. This RCT stated that PS placement using 3D fluoroscopy based on computed navigation was more accurate than the free-hand technique (PS misplacement rate: 2% versus 23%, *p* < 0.001). No PS-related complications were recorded in any of the patients. The clinical relevance of this study remains, however, unclear, as only 33 patients (478 PS) were included in total, which makes result generalization questionable, and as PS misplacements were defined by any cortical breach seen in routinely performed postoperative CT scans, which tends to overreport clinically irrelevant breaches, which might at least partly account for the significant accuracy difference found while comparing both techniques. The highest evidence supporting the superiority of computed navigation over the free-hand technique concerning PS placement accuracy is a meta-analysis published by Tian et al. in 2017 [41]. It included seven retrospective comparative series and the hereabove-described RCT and totaled 321 patients (3821 PS). The PS misplacement rate was significantly different among both groups, with 11.9% for the free-hand group and 3.7% for the navigation group. PS misplacement was defined as any breach over 2 mm seen in the routinely postoperatively performed CT scans. Among all patients, only one included in the free-hand group needed a revision surgery to remove a medially misplaced PS, causing late radiculopathy with leg weakness which resolved postoperatively. 

When referring to computed navigation, different technologies seem to achieve an equivalent PS placement accuracy, with reported PS misplacement rates between 1.9 and 11% in pediatric deformity patients. One relies on preoperative (prone or supine) CT scan performance with subsequent matching using intraoperative surface registration with a navigated probe which is localized by geometrical triangulation by two infrared cameras. Other technologies like 3D fluoroscopy, O-arm, and mobile multidetector CT rely on intraoperative image acquisition. All these intraoperative imaging technologies require a navigated clamp secured to a spinous process in the region of interest which is localized by two infrared cameras during image acquisition for matching. These latter technologies don’t need intraoperative surface registration, which has been shown to be time-consuming [35]. Kotani et al. compared the accuracy of O-arm-based navigation to preoperative CT scan performance with intraoperative surface registration in 61 pediatric deformity patients (638 PS) and found no significant difference (5% versus 3.1% misplaced PS, *p* > 0.05) [35]. Another comparative study by Zhang et al., including 67 pediatric deformity patients operated either using preoperative CT performance with intraoperative surface registration or intraoperative image acquisition with an O-arm or 3D fluoroscopy (Arcadis, Siemens, Muenchen, Germany), found similar results (5.6% versus 3.7% misplaced PS, *p* > 0.05) [42]. 

Robotic-assisted PS placement, which was introduced in 2004, has been reported to be more accurate than free-hand or computed navigation PS placement in adult spine surgery for treating various conditions [43]. For pediatric deformity surgery, some reports support the superiority of robotic-assisted PS placement, but no evidence has confirmed this statement until now. Case series of various sizes reported PS misplacement rates between 0.4% and 7.2% using robotic navigation for pediatric deformity surgery [19,37,38,39,40]. The PS-misplacement-related complications were not always reported and, when reported, were 0%. The study reporting the lowest PS misplacement rate (0.4%) is the largest available series (n = 162) but has a relevant limitation, as 87% of the cases were reviewed for PS misplacement only on the basis of fluoroscopic assessment which was not further described (no PS misplacement definition). Akazawa et al. recently published the first case-control study investigating the PS placement accuracy in 50 adolescent idiopathic scoliosis cases operated either using a preoperative CT-based navigation with intraoperative surface registration or robotic navigation based on preoperative CT with intraoperative fluoroscopic matching [19]. They found a higher PS misplacement rate for the CT navigation group in comparison to the robotic navigation group (7.5% versus 1.6%, *p* < 0.001).

Lastly, mere evidence is available concerning the PS placement accuracy using patient-specific drill templates with pre-planned trajectories, which are made of acrylate resin on the basis of fine-cut preoperative CT scans. Lu et al. reported 16 pediatric scoliosis patients (14 adolescent idiopathic scoliosis and 2 congenital scoliosis) operated using this technique [34]. Among the 168 implanted PS, 1.8% were considered as misplaced when defining any breach as a PS misplacement. When considering PS misplacement as a breach over 2 mm in any direction, no more PS misplacement was found. While this technology seems accurate and relatively cheap, its potential benefits still need to be confirmed by further research.

## 6. Comparison of Radiation Exposure According to the Used Surgical Technique

The cumulative radiation exposure of patients with adolescent idiopathic scoliosis is of concern. Indeed, Simony et al., reported a series of 209 adolescent idiopathic scoliosis cases treated with a Boston brace or posterior fusion with Harrington rod instrumentation between 1983 and 1990 in Denmark [44]. After 25 years of follow-up, these patients had a fivefold increase of cancer risk (4.3%) in comparison to the age-matched Danish population. Further, Luan et al. published a meta-analysis including 18,873 scoliosis patients and 16,768 controls as regional matched general population and found statistically higher cancer rates (any type) [OR = 1.46, *p* < 0.00001], breast cancer rates [OR = 1.2, *p* = 0.02], and cancer mortality rates [OR = 1.5, *p* < 000001] in the scoliosis patients in comparison with the controls [45]. Their mean number of full-spine X-rays performed during scoliosis follow-up was 23. 

Considering these facts, minimizing the radiation exposure of scoliosis patients is of utmost importance. When focusing on perioperative radiation exposure, it can be influenced by the chosen type and setting of the imaging used (conventional fluoroscopy, 3D fluoroscopy, O-arm, CT) as shown in Table 2. 

According to four publications, free-hand PS placement with conventional fluoroscopic control exposed the patients to a mean fluoroscopic time varying between 24 and 35 s, corresponding to an effective dose of 0.17 to 0.34 mSv [46,47,48,49]. When using computed navigation with an O-arm, the patient’s effective dose was reported to vary between 1.11 and 1.48 mSv, according to two case-control series. Su et al. reported an effective dose of 0.65 mSv for each low-dose O-arm scan, which was equivalent to 85 s of conventional fluoroscopy and allowed to visualize 6–8 vertebrae [49]. They also reported that when using the manufacturer’s recommended setting, each scan would expose the patients to an effective dose of 4.65 mSv. Sensakovic et al. developed a low-dose CT protocol for use in robot-assisted (Mazor^TM^, Medtronic, Dublin, Ireland) pediatric spinal surgery [51]. They reported an effective dose of 7.5 versus 0.95 mSv when using the standard, respectively, their low-dose protocol for preoperative spine imaging.

The surgeon’s usage of the available intraoperative imaging also largely influences the patient’s radiation exposure. When conventional fluoroscopy is used, the fluoroscopy time is subject to variations according to the surgeon’s technique and experience. When computed navigation is used (3D fluoroscopy, O-arm, or CT), the surgeon’s choice to perform, or not, a post-implantation scan to control PS positioning also largely influences the patient’s radiation exposure. In summary, robotic navigation and computed navigation induce a roughly fourfold increase in the patient’s intraoperative radiation exposure compared with conventional fluoroscopy. 

## 7. Conclusions

PS instrumentation and fusion for AIS and other pediatric deformity cases is a safe procedure, regardless of the used technique, as reported PS misplacement related neurologic, vascular, and visceral complication rates are low (0–1.4%). While some reports show superior PS placement accuracy of robotic navigation over computed navigation, and of the latter over the free-hand technique, related clinical benefits in complication reduction have not been demonstrated so far. Therefore, and because robotic navigation and computed navigation use induce a roughly fourfold increase in the patient’s intraoperative radiation exposure in comparison to the use of the free-hand technique with fluoroscopic implant positioning control, the authors do not recommend to use these modern technologies routinely for every pediatric deformity case, but selectively for spine deformities of high complexity and for revision cases with altered anatomical landmarks. The authors further highlight the need for a generally accepted PS accuracy classification to improve direct comparison of ongoing research and advise against PS placement in grade 4 pedicles yielding higher misplacement rates. 

## Figures and Tables

**Table 1 diagnostics-13-02402-t001:** This table compares PS placement accuracy for deformity correction in children using various placement techniques, misplacement definitions, and imaging modalities to assess PS position.

Author	Study Design	Number of Patients (PS)	PS Placement Technique	PS Misplacement Rate (Per Patient)	PS Accuracy Assessment	PS-Related Complications
Suk et al. [20]	Retrospective cohort	462 (4604)	free-hand	1.5% (10.4%) ^1^	X-ray-based, CT optional	0.2% ^2^
Li et al. [23]	Retrospective cohort	208 (1123)	free-hand	1.7% (7.2%) ^3^	X-ray-based, CT optional	1% ^4^
Modi et al. [24]	Retrospective cohort	43 (854)	free-hand	7% (na) ^5^	CT-based	0%
Modi et al. [25]	Retrospective cohort	26 (482)	free-hand	9.3% (na) ^6^	CT-based	0%
Kwan et al. [26]	Retrospective cohort	140 (2020)	free-hand	20.3% ^3^ or 2.2% ^7^ (na)	CT-based	0.1% ^8^
Liljenqvist et al. [27]	Prospective cohort	32 (120)	free-hand	25% (na) ^3^	CT-based	0%
Upendra et al. [28]	Prospective cohort	24 (138)	free-hand	50.7% ^3^ or 8.7% ^9^ (na)	CT-based	1.4% ^10^
Sakai et al. [29]	Retrospective case control	40 (478)	free-hand versus CT-nav. ^11^	28% versus 11% (na) ^12^, *p* < 0.05	CT-based	0%
Cui et al. [30]	Retrospective case control	59 (1040)	free-hand versus CT-nav. ^13^	5.2% versus 1.9% (na) ^3^, *p* < 0.05	CT-based	0% versus 0%
Ughwanogho et al. [31]	Retrospective case control	42 (485)	free-hand versus O-arm nav.	9% versus 3% ^14^ (na), *p* < 0.001	O-arm-based	0% versus 0%
Rajasekaran et al. [32]	Randomized controlled trial	33 (478)	free-hand versus 3D-fluoroscopy nav.	23% versus 2% ^3^ (na), *p* < 0.001	CT-based	0% versus 0%
Larson et al. [33]	Retrospective cohort	50 (984)	O-arm nav.	3.6% (na) ^15^	CT-based	0%
Lu et al. [34]	Prospective cohort	16 (168)	drill templates	1.8% ^3^ versus 0% ^12^ (na)	CT-based	0%
Sarwahi et al. [11]	Retrospective cohort	127 (2724)	na	1.1% (14.2%) ^16^	CT-based	0%
Kotani et al. [35]	Retrospective case control	61 (638)	O-arm versus CT nav. ^11^	5% versus 3.1% (na) ^15^, *p* > 0.05	CT-based	0% versus 0%
Zhang et al. [36]	Retrospective case control	67 (1118)	pre versus intra-operative nav. ^17^	5.6% versus 3.7% ^12^ (na), *p* > 0.05	CT-based	0%
Gonzalez et al. [37]	Retrospective cohort	40 (314)	robotic nav. ^18^	1.3% (na) ^19^	Fluoroscopy & X-ray-based	0%
Macke et al. [38]	Retrospective cohort	50 (662)	robotic nav. ^20^	7.2% ^12, 21^ or 2.4% ^12, 22^ (na)	CT-based	na
Welch et al. [39]	Retrospective cohort	162 (1467)	robotic nav. ^23^	0.4% ^24^ (na)	fluoroscopy-based, CT optional ^24^	0%
Sawires et al. [40]	Retrospective cohort	14 (95)	robotic nav. ^25^	1% (7%) ^26^	fluoroscopy-based, CT optional ^26^	0%
Akazawa et al. [19]	Retrospective case control	50 (741)	CT ^11^ versus robotic nav. ^20^	7.5% versus 1.6% ^12^ (na), *p* < 0.001	CT-based	na
Ledonio et al. [7]	Systematic review	1353 (13,536)	na	5.1% (na) ^15^	na ^27^	na ^27^
Hicks et al. [21]	Systematic review	1666 (12,248)	free-hand but 1 study 3D-fluoroscopy nav.	4.2% (around 10%) ^28^	X-ray-based or CT-based	0.1% ^2^
Tian et al. [41]	Meta-analysis (8 papers)	321 (3821)	free-hand versus nav.	11.9% versus 3.7% (na) ^12^	CT-based	na ^27^

“nav.” is an abbreviation for navigation. “na” is an abbreviation for not available. ^1^ PS misplacement qualitatively defined (medial, lateral, superior, inferior); ^2^ one transient paraparesis; ^3^ any breach; ^4^ one patient with neurologic complication had a transient paraparesis due to a misplaced PS which dislocated more medially during the first postoperative month; ^5^ significant misplacement, considered as any medial breach respective lateral breaches over 2 mm; ^6^ PS misplacement was defined as medial breaches over 2 mm and breaches over 4 mm in any other direction; ^7^ critical PS misplacement defined as medial or lateral breaches of at least 2 mm, or anterior breaches of at least 4 mm, while lateral breaches in the thoracic region were not considered as critical PS misplacements, as those PS were contained in the pedicle-rib junction; ^8^ one incomplete dermatomal numbness with partial improvement during FU and one painful radiculopathy which resolved under conservative treatment; ^9^ PS misplacement was defined by a medial breach of at least 2 mm, a lateral breach outside the pedicle-rib unit, or any breach with documented (CT or MRI) PS-misplacement-related neurovascular complication; ^10^ two patients with transient neurologic deficit; ^11^ preoperative CT scan performance with intraoperative surface registration for matching; ^12^ breaches over 2 mm, any direction; ^13^ intraoperative CT scan performance; ^14^ PS misplacement was defined as a PS with its central axis crossing the midline of the corresponding vertebral body, respectively a PS breaching anterolaterally and placing the aorta at risk or any PS repositioned or removed based on the intraoperative O-arm-based imaging; ^15^ PS misplacement not defined; ^16^ Screw at risk according to Sarwahi et al. [11]; ^17^ preoperative CT performance with intraoperative surface registration versus intraoperative O-arm or 3D-fluoroscopy based navigation; ^18^ preoperative CT was merged with intraoperative fluoroscopic images in 12 patients, and 3D acquisition was obtained intraoperatively using an O-arm in 28 patients; ^19^ breaches of at least 2 mm and under 4 mm, any direction; ^20^ preoperative CT was merged with intraoperative fluoroscopic images in all patients; ^21^ supine preoperative CT; ^22^ prone preoperative CT; ^23^ preoperative CT was merged with intraoperative fluoroscopic images in 31 patients, and 3D acquisition was obtained intraoperatively using an O-arm in 131 patients; ^24^ PS misplacement was not defined for fluoroscopic PS placement assessment. 13% of the PS were assessed with CT and were completely contained in the corresponding vertebra; ^25^ preoperative CT was merged with intraoperative fluoroscopic images in 4 patients, and 3D acquisition was obtained intraoperatively using an O-arm in 10 patients; ^26^ 55% of the screws were assessed with CT and the others with fluoroscopy. Those assessed with CT were graded according to the Gertzbein-Robbins classification [10]; ^27^ Not systematically reported; ^28^ one study defined PS misplacement as a breach over 2 mm and the 11 others as any breach.

**Table 2 diagnostics-13-02402-t002:** Table comparing PS placement associated patients’ effective dose using the free-hand technique with fluoroscopic implant positioning control to the use of computed navigation.

Author	Diagnosis	Study Design	Number of Patients	Imaging Type	Patient Effective Dose (mSv)	Average Levels Fused
O’Donnell et al. [46]	AIS	Retrospective case series	43	Fluoroscopy	0.189 mSv/26 s	11
Berlin et al. [47]	AIS	Retrospective case series	73	Fluoroscopy	0.17 mSv/24 s	9.5
Dabaghi et al. [48]	Pediatric deformity	Prospective case control series	81	Fluoroscopy versus O-arm	0.34 versus 1.48 mSv, *p* = 0.0012	na
Su et al. [49]	Pediatric scoliosis	Retrospective case control series	28	Fluoroscopy versus O-arm	0.27 versus 1.11 mSv ^a^	11 ^b^
Urbanski et al. [50]	AIS	Prospective case control series	49	Fluoroscopy versus O-arm	391 versus 1071 mGy-cm; *p* > 0.001 ^d^	11.7 versus 11.3
Sensakovic et al. [51]	Pediatric scoliosis	Retrospective case control series	34	Standard versus low-dose CT	Preop CT 7.5 versus 0.95 mSv. Post op CT 10.5 versus 0.89 mSv ^c^	na

“AIS” is an abbreviation for adolescent idiopathic scoliosis and “na” for not available. ^a^ One to two O-arm scans were done per patient. The average fluoroscopy time (mean ± SD) was 35.3 ± 23.9 s (range, 7.9 to 75.0 s) corresponding approximately to 3 s per level fused; ^b^ The median number of imaged levels was 11 (PS density of 1.3 to 1.7 per level fused); ^c^ scan length 7.2 cm longer for low-dose cohort; ^d^ Radiation exposure expressed in mGy-cm (dose-length product) instead of patients’ effective dose.

## Data Availability

Not applicable.

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
