# Peer review of "Accuracy and Safety of Pedicle Screw Placement for Treating Adolescent Idiopathic Scoliosis: A Narrative Review Comparing Available Techniques"

_diagnostics, 2023, doi:10.3390/diagnostics13142402_

Round 1

Reviewer 1 Report

First of all, I would like to congratulate you on the successful progress of your useful review study.

This study was a well-reviewed study of computer navigation, robotic navigation, patient-tailored drill templates, and freestyle techniques used to reduce errors when inserting pediatric screw (PS) instruments to correct adolescent idiopathic scoliosis. This paper compares PS placement accuracy with the ratio of complications associated with placement errors when each technique is applied, and provides useful information to readers by summarizing PS accuracy classification and anatomical placement risk factors well. Furthermore, a comprehensive review of the benefits of using robots, the need for calculated retrieval, and the risk of increased intraoperative radiation exposure for complex deformations with altered landmarks has been well performed.

So this paper can be published as it is.

Author Response

Dear reviewer,

Thank you very much for your positive comments.

============================================

Reviewer 2 Report

The authors have successfully compared different techniques for pedicle screw placement.

The manuscript is well written. The literature data have been thoroughly analyzed and presented. The paper gives valuable information for pedicle screw instrumentation in scoliosis surgery. I believe it will appeal to the readers of the journal and therefore I suggest it be published.

Author Response

(The authors gave the same response as above.)

Reviewer 3 Report

Thank you for submitting your review article on the accuracy and safety on the use of pedicle screws in adolescent idiopathic scoliosis (AIS). It was a timely, well-written review. I have only few suggestions for a revision, if requested.

Abstract

(1). Page 1, Line 12. Please remember to revise your abstract after your revision to incorporate your changes.

(2). Page 1, Line15. This sentence is repetitive and needs to be revised.

Introduction

(3). Page 1, Line 31. Your opening sentence is confusing. You use this same phrase elsewhere in your manuscript. Please revise to something  similar to "posterior spinal fusion (PSF) and segmental spinal instrumentation (SSI) using pedicle screws through". This more accurately describes what you are trying to convey.

(4). Page 2, Line 48. A major concern is the presentation of a large number of patients or pedicle screws was presented using an apostrophe rather than a coma (13' 536 vs 13,536 pedicle screws). I am not aware if the former is commonly used in Europe rather than other countries. In fact, I have never seen it used in scientific writing previously. I would suggest this be changed to the latter.

PS Placement Accuracy Classifications

(5). Page 3, Line 103. Is "totalizing" a real word. Please consider revising.

Patient-Related Risk-Factors for Screw Misplacement

(6). Page 3, Line 137. This sentence would be better expressed if the total number of pedicle screws was stated as well as the number in each classification. This would also justify the use of a single decimal point in percentages.

(7). Page 4, Line 165. The same concern here as in Question #6.

PS Misplacement-Related Morbidity and Mortality

(8). Page 4, Line 165. Please use only one decimal place in expressing percentages. Make this change throughout your manuscript,  including any tables or figure legends. 

Conclusions

(9). Page 9, Line 391. Excellent conclusion. Your study was accurate and very informative.

Illustrations

(10). Consider adding illustrations demonstrating the different classifications of pedicle anatomy. I think this would be interest to our readers. Your decision.

English needs a minor review to make it more scientific. My major concern is the presentation of a large number of patients or pedicle screws was presented using an apostrophe rather than a coma. I am not aware this is commonly used in Europe rather than other countries. In fact I have never seen it used in scientific writing previously.

Author Response

Comments and Suggestions for Authors

Thank you for submitting your review article on the accuracy and safety on the use of pedicle screws in adolescent idiopathic scoliosis (AIS). It was a timely, well-written review. I have only few suggestions for a revision, if requested.

Abstract

(1). Page 1, Line 12. Please remember to revise your abstract after your revision to incorporate your changes.

Answer to comment nr. 1: Thank you for this comment. We revised the abstract taking into account this comment and your comment nr. 3.

(2). Page 1, Line15. This sentence is repetitive and needs to be revised.

Answer to comment nr. 2: Thank you for this pertinent comment. We revised the sentence as follows to be more clear: This review compares the PS placement accuracy and misplacement-related complication rates achieved with the techniques mentioned above. It further reports PS accuracy classifications and anatomic PS misplacement risk factors.

Introduction

(3). Page 1, Line 31. Your opening sentence is confusing. You use this same phrase elsewhere in your manuscript. Please revise to something similar to "posterior spinal fusion (PSF) and segmental spinal instrumentation (SSI) using pedicle screws through". This more accurately describes what you are trying to convey.

Answer to comment nr. 3: Thank you for this comment. We revised the opening sentence of the introduction according to your comment and also revised the abstract where the same phrase was used in the first manuscript.

(4). Page 2, Line 48. A major concern is the presentation of a large number of patients or pedicle screws was presented using an apostrophe rather than a coma (13' 536 vs 13,536 pedicle screws). I am not aware if the former is commonly used in Europe rather than other countries. In fact, I have never seen it used in scientific writing previously. I would suggest this be changed to the latter.

Answer to comment nr. 4: Thank you for this constructive comment. We revised the whole manuscript according it.

PS Placement Accuracy Classifications

(5). Page 3, Line 103. Is "totalizing" a real word. Please consider revising.

Answer to comment nr. 5: Thank you for this comment. We replaced the word "totalizing" by "summarizing".

Patient-Related Risk-Factors for Screw Misplacement

(6). Page 3, Line 137. This sentence would be better expressed if the total number of pedicle screws was stated as well as the number in each classification. This would also justify the use of a single decimal point in percentages.

Answer to comment nr. 6: Thank you for this constructive comment. We revised the manuscript according to it as follows: The distribution of pedicle channel types varied according to the diagnosis leading to scoliosis and was as follows for the included 38 AIS cases operated using 474 PS: 254 PS (53.6%) Type A, 157 PS (33.1% )Type B, 39 PS (8.2%) Type C and 24 PS (5.1%) Type D.

(7). Page 4, Line 165. The same concern here as in Question #6.

Answer to comment nr. 7: Thank you for this pertinent comment. The cited study didn’t reported the number of failures for each pedicle channel grade but only the percentage. The number of probed pedicle was available as well as the number of failures in total. We therefore made the following changes in the revised manuscript: There were 61 failures among the 810 probed pedicle channels. The failure rate breakdown was as follows for grades 1 to 4: 0.5%, 2.9%, 12% and 31.5% (p<0.001).

PS Misplacement-Related Morbidity and Mortality

(8). Page 4, Line 165. Please use only one decimal place in expressing percentages. Make this change throughout your manuscript, including any tables or figure legends.

Answer to comment nr. 8: Thank you for this comment. We made the necessary changes throughout the whole manuscript.

Conclusions

(9). Page 9, Line 391. Excellent conclusion. Your study was accurate and very informative.

Answer to comment nr. 9: Thank you for this positive comment.

Illustrations

(10). Consider adding illustrations demonstrating the different classifications of pedicle anatomy. I think this would be interest to our readers. Your decision.

Answer to comment nr. 10: We agree with this comment. If an illustration is requested we will ask for copyright clearance of the figure 1 of ref. nr. 18 by Akazawa et al.

Comments on the Quality of English Language

English needs a minor review to make it more scientific. My major concern is the presentation of a large number of patients or pedicle screws was presented using an apostrophe rather than a coma. I am not aware this is commonly used in Europe rather than other countries. In fact I have never seen it used in scientific writing previously.

Answer to the comment on the quality of English language: Thank you for this comment. We
made changes throughout the revised manuscript according to it.